# Antioxidant Activity of Phenolic Extraction from Different Sweetpotato (*Ipomoea batatas* (L.) Lam.) Blades and Comparative Transcriptome Analysis Reveals Differentially Expressed Genes of Phenolic Metabolism in Two Genotypes

**DOI:** 10.3390/genes13061078

**Published:** 2022-06-16

**Authors:** Peitao Chen, Hairong Ran, Jiaxin Li, Jikai Zong, Qingqing Luo, Tengfei Zhao, Zhihua Liao, Yueli Tang, Yufan Fu

**Affiliations:** Engineering and Technology Research Center for Sweetpotato of Chongqing, School of Life Science, Southwest University, Chongqing 400715, China; capital0529@163.com (P.C.); rhr19970712@163.com (H.R.); jiaxinli0806@163.com (J.L.); zongjikai@126.com (J.Z.); yxcf99ss@163.com (Q.L.); tengfeizhao@swu.edu.cn (T.Z.); zhliao@swu.edu.cn (Z.L.); tangyueli@swu.edu.cn (Y.T.)

**Keywords:** sweetpotato blade, antioxidant activity, phenolic substances, differentially expressed genes, phenylpropanoid biosynthesis pathway

## Abstract

Sweetpotato (*Ipomoea batatas* (L.) Lam.), which has a complex genome, is one of the most important storage root crops in the world. Sweetpotato blades are considered as a potential source of natural antioxidants owing to their high phenolic content with powerful free radical scavenging ability. The molecular mechanism of phenolic metabolism in sweetpotato blades has been seldom reported thus far. In this work, 23 sweetpotato genotypes were used for the analysis of their antioxidant activity, total polyphenol content (TPC) and total flavonoid content (TFC). ‘Shangshu19’ and ‘Wan1314-6’ were used for RNA-seq. The results showed that antioxidant activity, TPC and TFC of 23 genotypes had significant difference. There was a significant positive correlation between TPC, TFC and antioxidant activity. The RNA-seq analysis results of two genotypes, ‘Shangshu19’ and ‘Wan1314-6’, which had significant differences in antioxidant activity, TPC and TFC, showed that there were 7810 differentially expressed genes (DEGs) between the two genotypes. Phenylpropanoid biosynthesis was the main differential pathway, and upregulated genes were mainly annotated to chlorogenic acid, flavonoid and lignin biosynthesis pathways. Our results establish a theoretical and practical basis for sweetpotato breeding with antioxidant activity and phenolics in the blades and provide a theoretical basis for the study of phenolic metabolism engineering in sweetpotato blade.

## 1. Introduction

Sweetpotato, an annual or perennial herb of the *Ipomoea* in Convolvulaceae, is a storage root crop. Because of its excellent agronomic characteristics such as drought tolerance, high photosynthetic efficiency and high yield [1,2], it is the fifth major food crop in developing countries ranked after rice, wheat, maize and potato. China is the largest producer of sweetpotato, with a planting area of 4.0 × 10^6^ hm^2^ and an annual production of 5.2 × 10^8^ t in 2020 [3].

Some studies have shown that sweetpotato stems, petioles and blades are rich in proteins, vitamins, minerals, polyphenols, flavonoids, anthocyanins and some functional trace components [4,5]. It not only has high nutritional value, but also has a variety of physiological health functions such as anti-cancer and anti-bacterial effects, improving immunity, preventing cardiovascular disease and diabetes [6,7,8], as well as has the effect of improving immunity for livestock [9]. Therefore, sweetpotato stems, petioles and blades have good health value for both humans and livestock, and they deserve our attention and exploitation. However, sweetpotato is planted mainly for harvesting its storage root, and only a small portion of stems, petioles and blades are used as forage and for human consumption, resulting in a great waste of resources. The estimated total production of sweetpotato stems, petioles and blades in 26 Chinese provinces in 2020, by normal T/R (ratio of top yield namely stem, petioles and blades yield to root yield) value as 0.4 at sweetpotato harvesting would reach at least 2.0 × 10^7^ t. Therefore, the resources for sweetpotato stems, petioles and blades are quite abundant.

Sweetpotato blade extracts have outstanding antioxidant activity. It was found that the crude extract of polyphenols in sweetpotato blades had strong free radical scavenging activity, reducing activity, metal chelating activity and liposome oxidative damage inhibiting activity [5,10]. Islam et al. characterized the content and types of polyphenols in sweetpotato blades of several genotypes and found that the polyphenol content in sweetpotato blades was much higher than that in other vegetables available in the market, and the chlorogenic acid types were especially abundant [5]. Ojong et al. determined methanolic extracts from nine genotypes of sweetpotato blades and storage roots, and the result showed that there are a high content and variety of flavonoid in blades [11]. Sun et al. found that the blades extract of ‘Yuzi7’ has excellent ·O^2−^ scavenging ability. The ·O^2−^ scavenging activity of 20 ug/mL ‘Yuzi 7’ blades extract was 3.1, 5.9 and 9.6 times of 20 ug/mL ascorbic acid, 20 ug/mL tea polyphenols and 20 ug/mL grape seed polyphenols, respectively [12]. These studies indicated that the antioxidant activity of sweetpotato blades extract is mainly related to phenolic compounds, while blades, which generally account for about 50% of the sweetpotato vines biomass [13], are the main contributors to the antioxidant activity of sweetpotato vines [14]. Sweetpotato blades are also easy to process and exploit due to their low lignin and cellulose content.

Antioxidant activity in plants is related to the phenols metabolism. For example, antioxidant activity, polyphenol and flavonoid contents increased after ABA (abscisic acid) treatment, and some genes of the phenylpropanoid pathway were upregulated by 1.13 to 26.95 times in tomato [15]. The accumulation of anthocyanin, a sort of flavonoid, in tomato leaves, was significantly increased under low temperature induction; simultaneously, related gene expression was also significantly increased [16]. When different lettuce genotypes were faced with Al-induced oxidative stress, the phenolic acids in those tolerant genotypes were significantly increased, and their contents of four phenolic acids and two flavonoids were also increased [17]. Physiological and transcriptomic studies revealed that the antioxidant activity of injured carrots increased, and the total soluble phenol content increased [18]. These studies suggest a correlation between phenolic metabolism and antioxidant activity. Transcriptome studies on polyphenol metabolism in sweetpotato blades has not been reported.

In this study, we measured and compared the differences of antioxidant activities, total polyphenol content and total flavonoid content of methanolic extracts from 23 genotypes of sweetpotato blades and used RNA-seq to analyze the differences of gene transcript levels in blades between two genotypes with significant differences in antioxidant activity and phenolic content. Our study provided a theoretical and practical basis for the selection and breeding of sweetpotato varieties with high phenolic content and antioxidant activity in its blade, as well as providing theoretical support for engineering research on phenolic metabolism of sweetpotato blades.

## 2. Materials and Methods

### 2.1. Plant Materials

Twenty-three sweetpotato genotypes (Appendix A) were provided by ‘Chongqing Engineering Research Center for Sweetpotato’ Southwest University China. These materials were grown on the field experimental base (30°0′32.79″ N 106°07′57.60″ E) of this research center in May 2021, and their blades were sampled 90 days after transplanting. Mature blades (4th to 6th blades from the stem apex), which were free of diseases and pests and fully expanded, were sampled from each genotype. Sampled blades were washed with distilled water, ground to powder in liquid nitrogen and stored at −80 °C.

### 2.2. Determination of Antioxidant Activity of Blade

ABTS+ (2,2′-azinobis-(3-ethylbenzothiazoline-6-sulphonic acid) free radicals), FRAP (ferric ion-reducing antioxidant power) and DPPH· (2,2-diphenyl-1-picrylhydrazyl) methods were used to describe the antioxidant activity of the samples. Methanolic extracts of the sample powders were prepared for backup, referring to Grimalt’s method [19]. ABTS+ scavenging assay was carried out as described by the method of Re et al. [20]. Briefly, 0.4 mL of the extract was mixed with 3.6 mL of ABTS+ solution, and the reaction was carried out at room temperature for 20 min, and the absorbance value was measured at 734 nm. FRAP assay was carried out as described by the method of Benzie et al. [21]. Briefly, 0.4 mL of extract was mix with 3.0 mL of FRAP reagent, reacted for 10 min at room temperature, and the absorbance at 593 nm was measured. Meanwhile, DPPH· scavenging capacity assay was carried out as described by the method of Chen et al. [22]. Briefly, 2.0 mL of the extract was mixed with 2.0 mL of DPPH· solution, with the reaction at room temperature and light avoidance for 30 min. The absorbance value was measured at 517 nm. The results of antioxidant activities were expressed as Trolox (6-hydroxy-2,5,7,8-tetramethylchroman-2-carboxylic acid) equivalent (mg TE/g).

### 2.3. Determination of Total Polyphenol Content and Total Flavonoid Content

Total polyphenol content (TPC) was determined referring to the Folin–Ciocalteu method [23]. Briefly, 1.0 mL of methanolic extract was drawn into a 10.0 mL volumetric flask, and then 5.0 mL of 10% Folin–Ciocalteu solution was added to the flask, shaken well and mixed with 4.0 mL of 7.5% Na_2_CO_3_ solution, placed in the dark for 30 min, and the absorbance value was measured at 765 nm. Calibration curves were plotted with gallic acid, and the TPCs were expressed as gallic acid equivalents (mg GAE/g).

Determination of total flavonoid content (TFC) was in accordance to Nie et al. [24]. Briefly, 0.5 mL of methanolic extract was pipetted into a centrifuge tube, and 1.0 mL of 0.05 g/mL NaNO_2_ was added, after 6 min, 1.0 mL of 0.1 g/mL Al(NO_3_)_3_ was added, and then 3.0 mL of 1.0 mol/L NaOH was added after 6 min, and the absorbance value was measured at 510 nm after a water bath at 25 °C for 15 min. Calibration curve was plotted with rutin, and the TFCs were expressed as rutin equivalents (mg RE/g).

### 2.4. cDNA Library Construction and RNA-Seq

Three biological replicates of each genotype were selected from the low antioxidant activity genotype ’Shangshu19’ and the high antioxidant activity genotype ‘Wan1314-6’ blades. Total RNA was extracted using the total RNA Extraction Kit (DP419, TIANGEN, Beijing, China), RNA concentration was measured using the Nanodrop (IMPLEN, Westlake Village, CA, USA), and RNA integrity and purity were accurately assessed using the Aligent 5400 (Agilent Technologies, Santa Clara, CA, USA). After the total RNA samples passed the test, the cDNA libraries were constructed and quality controlled by Novogene Co., Ltd. (Beijing, China). The constructed libraries were sequenced with Illumina HiSeqTM platform for transcriptome sequencing by Novogene Co., Ltd.

### 2.5. Transcriptome Assembly and Functional Annotation

Raw reads were filtered by Qphred ≥ 20 and read length was ≥ 25 bp using SolexaQA package to remove reads with an adapter, which also removed N-containing reads, and removed low-quality reads (reads with Qphred ≤ 20 bases accounting for more than 50% of the entire read length). In addition, Q20, Q30 and GC content calculations were performed on the clean data. All subsequent analyses were performed based on clean data with high quality. Mapping clean reads to the sweetpotato reference genome used HISAT2 v2.0.5. (http://public-genomes-ngs.molgen.mpg.de/Sweet potato, (accessed on 1 November 2021)).

### 2.6. Differentially Expressed Genes (DEGs) Analysis

FPKM (fragments per kilobase of exon model per million mapped fragments) was used to represent gene expression levels, and DESeq (1.20.0) was used for screening of DEGs (screening criteria were |log2FC| ≥ 2, padj < 0.05, FDR < 0.01). GO (Gene Ontology) enrichment analysis and KEGG (Kyoto Encyclopedia of Genes and Genomes) pathway enrichment analysis used topGO R and KOBAS (V2.0) package (Significant enrichment criterion were corrected at *p* value < 0.05).

### 2.7. Quantitative Real-Time PCR (qRT-PCR) Analysis of DEGs

To verify the reliability of the transcriptome results, 10 genes from the DEGs were selected for qRT-PCR. FastQuant RT Kit (TIANGEN, Beijing, China) was used to synthesize first-strand cDNA with gene-specific primers (Appendix A). The *IbACTIN* gene was used as the reference, and NovoStar^®^SYBR qPCR SuperMix Plus (Novoprotein, Shanghai, China) was used for PCR amplification and DNA staining. qRT-PCR was performed on an IQ5 thermal cycler (Bio-Rad, Hercules, CA, USA). Relative gene expression levels were calculated according to the 2^−ΔΔCT^ method [25].

### 2.8. Statistical Analysis

Mean and standard error of biological replicates were calculated using Microsoft Excel 2016; SPSS software (IBM, Armonk, NY, USA) was used for analysis of variance (ANOVA) and correlation analysis at *p* ≤ 0.05. TBtools [26] was used for heatmap plotting.

## 3. Results

### 3.1. Antioxidant Activity of Sweetpotato Blades

The results of ABTS+, DPPH· and FRAP antioxidant activity of blades from 23 sweetpotato genotypes were visualized by heatmap (Figure 1). The ABTS+ scavenging activities ranged from 4.59 to 27.13 mg TE/g, and ‘161,614’ had the highest value (27.13 mg TE/g), which was 5.9 times higher than that of the lowest from ‘Zhongshu1’ (4.59 mg TE/g). The DPPH· scavenging activities ranged from 3.93 to 20.76 mg TE/g, and ’18-2-21’ had the highest value (20.76 mg TE/g), which was 5.3 times higher than that of the lowest from ‘Shangshu19’ (3.93 mg TE/g). The FRAP scavenging activities ranged from 9.64 to 85.52 mg TE/g, and ‘Wan1314-6’ had the highest value (85.52 mg TE/g), which was 8.9 times higher than that of the lowest from ‘Shangshu19’ (9.64 mg TE/g). In summary, the three antioxidant activity indicators had an obvious parallel relationship, and the antioxidant activities were significantly different among the 23 genotypes (Appendix A).

### 3.2. TPC and TFC of Sweetpotato Blades

Phenolic compounds may play an important role in the antioxidant activity of plants. The TPC and TFC of sweetpotato blades were determined for 23 genotypes, and a heatmap was used for visualization (Figure 2). The TPC of the 23 genotypes varied from 2.17 to 28.32 mg GAE/g, with significant differences among genotypes (Appendix A). The highest TPC of ‘Wan1314-6’ was 28.32 mg GAE/g, the TPC content of Shangshu19 was 10.08 mg GAE/g, while the lowest TPC of ‘18-6-24’ was only 2.17 mg GAE/g. There were significant differences in TFC among the 23 genotypes, ranging from 3.07 to 10.73 mg RE/g. Genotype ‘Wan1314-6’ contained the highest TFC of 10.73 mg RE/g, ‘18-6-43’ had the lowest TFC of 3.07 mg RE/g, ‘Zhongshu1’ contained 3.58 mg RE/g, ‘Shangshu19’ contained 4.44 mg RE/g, and there were no significant differences between them. Heatmap showed a clear parallel relationship between TPC and TFC content.

### 3.3. Correlation Analysis between Antioxidant Activity, TPC and TFC

To investigate whether there was a correlation between the antioxidant activity, TPC and TFC of sweetpotato blades, the correlation analysis results among ABTS+, DPPH·, FRAP, TPC and TFC are shown in Figure 3.

All five indicators had significant positive correlations with each other, among which the correlation coefficients between the three antioxidant activities were 0.58, 0.83 and 0.53, respectively, which proved that the simultaneous use of three methods to assess the antioxidant activity of the samples was reliable; the flavonoids belonged to phenolic compounds, and the correlation coefficient between TPC and TFC was 0.85. The correlation coefficients of TPC with the three antioxidant indicators were 0.75, 0.84, and 0.83, respectively, while the correlation coefficients of TFC with the three antioxidant indicators were 0.68, 0.58, and 0.90, respectively.

### 3.4. RNA-Seq and Data Pre-Processing

Two genotypes, ‘Shangshu19’ (S) and ‘Wanshu1314-6’ (W) blades, were selected for RNA-seq. Three libraries were created for RNA-seq for each genotype, and the correlation coefficient between different replicates for each genotype was greater than or equal to 0.955 (Figure 4), in accordance with the Encode program recommendations (R2 ≥ 0.92).

According to the RNA-seq results (Table 1), the average base sequencing error rate of the library sequencing data was around 0.03; the percentage of bases with Phred values greater than 20 in the overall bases ranged from 96.99% to 97.67%, and the percentage of bases greater than 30 in the overall bases ranged from 92.00% to 93.64%. The results showed high sequencing quality, and most clean reads were retained for further analysis.

### 3.5. Analysis of DEGs

The DEGs of W and S were analyzed using DESeq2 (1.20.0), and the volcano plot (Figure 5A) clearly shows the distribution of differential genes, and there was a large number of differentially expressed genes in W and S. As shown by the column figure (Figure 5B), there were 7810 DEGs in W and S groups, and compared with S, W had 3968 upregulated genes and 3842 downregulated genes.

Gene Ontology (GO) is an international standardized gene functional classification system. To further evaluate the functions of DEGs, this study used GO to functionally classify the sweetpotato W and S blades transcripts. The results showed (Figure 6A) that a total of 31,999 transcripts were mapped to different GO function nodes, and the 30 most significant terms were selected for scatter plotting, with 10,912 transcripts attributed to ‘biological process’, 3278 transcripts attributed to ‘cellular component’ and 17,809 transcripts attributed to ‘molecular function’, respectively. The highest proportion of transcripts involved in the ‘multi-organism process’ was found in BP, followed by ‘cell recognition’ and ‘multi-multicellular organism process’. In CC, the proportion of ‘extracellular region’, ‘apoplast’, and ‘cell wall’ were the top three.

We performed KEGG pathway analysis on the DEGs. KEGG pathway analysis can systematically analyze the metabolic pathways of gene products in cells, and the results showed that 12,924 transcripts were annotated to 131 KEGG pathways, and the 20 most significant KEGG pathways were selected to plot scatter plots, as shown in Figure 6B. The DEGs were significantly enriched in ‘phenylpropanoid biosynthesis’, ‘cytochrome P450’, ‘amino sugar and nucleotide sugar metabolism’, ‘carotenoid biosynthesis’, ‘glutathione metabolism’ and other pathways, and the number of differential genes was 75, 44, 38, 18 and 24, respectively.

KEGG enrichment analysis showed that the genotypes W and S blades differed significantly in their phenylpropanoid biosynthesis pathway, which is the main pathway of polyphenol origin and the upstream pathway of flavonoid compound synthesis. After further analysis of upregulated genes in the phenylpropanoid biosynthesis pathway and flavonoid biosynthesis pathway, a total of 30 upregulated expressed sequences were annotated to 15 key enzymes (Appendix A) after NCBI (National Center for Biotechnology Information Search database) database comparison. The metabolic pathway of phenylpropanoid biosynthesis is shown in Figure 7A, and the significantly upregulated genes were mainly key genes for the synthesis of phenolic acid compounds. For example, *PAL* (phenylalanine ammonia lyase), *4CL* (4-Coumarate:coenzyme A ligase), *CCoAOMT* (caffeoyl-CoAO-methyltransferase), *POD* (peroxidase), *CAD* (cinnamyl-alcohol dehydrogenase), *F5’H* (ferulic acid 5-hydroxylase), *HCT* (hydroxycinnamoyl-CoA shikimate), *ALDH2C4* (aldehyde dehydrogenase 2C4). The metabolic pathway of flavonoid biosynthesis is shown in Figure 7B. The enzymes significantly upregulated in the pathway were mainly *CHS* (chalcone synthase), *CHI* (chalcone isomerase), and *F3’H* (flavanone 3-hydroxylase). These upregulated genes were mainly annotated in the chlorogenic acid, flavonoid and lignin synthesis pathways.

### 3.6. qRT-PCR Verification of the Phenylpropanoid Biosynthesis Pathway

To verify the reliability of the transcriptome data, 10 of the upregulated DEGs of the phenylpropanoid biosynthetic pathway were randomly selected for qRT-PCR analysis. Ten DEGs were expressed at higher levels than S in W (Figure 8A). Correlation analysis showed that there was a significant positive correlation between RNA-seq and qRT-PCR (Figure 8B).

## 4. Discussion

Sweetpotato vines have important processing, edible and potential medicinal values, and are receiving more and more attention in their industrialization in southern China [27].

Plant tissues contain a number of natural antioxidants. The antioxidant activity of sweetpotato is especially prominent, and some sweetpotato varieties have antioxidant activity even stronger than blueberries [28]. The antioxidant activity of different parts of sweetpotato varies greatly, and the antioxidant activity of its blades is usually greater than that of other parts. Thus, sweetpotato blades are also regarded as a potential natural antioxidant [29]. The most important chemical components of sweetpotato blades are polyphenolics and flavonoids [5]. In this study, antioxidant activity TPC and TFC of sweetpotato blades were determined for 23 genotypes, and the variation in TPC in 23 genotypes blades ranged from 2.17 to 28.32 mg GAE/g, and TFC ranged from 3.58 to 10.95 mg RE/g. The results were similar to some reports [30,31,32,33]. The TPC and TFC of some genotypes blades with high antioxidant activity were also high; for example, genotype ‘Wan1314-6’ with high antioxidant activity ranked first in TPC and second in TFC among the 23 genotypes, while genotypes ‘Zhongshu1’ and ‘Shangshu19’ with low antioxidant activity ranked low in TPC and TFC among the 23 genotypes. The results of the correlation analysis of ABTS+, DPPH·, FRAP, TFC and TPC also indicated that phenolic compounds have an extremely important role in conferring antioxidant activity to sweetpotato blades. Since sweetpotato is a hexaploid species with a large and complex genome, breeding objectives have an important guiding role in the selection of parents in the sweetpotato breeding progress [34]. In our study, ‘Wan1314-6’, ‘Yushu1’ and ‘18-2-4’ with high contents of TPC and TFC and high antioxidant activity would be suitable as parents for breeding sweetpotato varieties with high antioxidant activity in their blades. Considering that plant traits are influenced by both genotypes and environmental factors (G × E), we will subsequently verify the stability of these genotypes through comparative experiments at different sites.

In order to further reveal the relationship between antioxidant mechanisms and phenolic compounds in sweetpotato blades, we analyzed the phenolic metabolism of two genotypes of sweetpotato blades with different TPC, TFC and antioxidant activities by comparing transcriptomes. Analysis of transcriptomic data showed that metabolic pathway differences were mainly localized to the phenylpropanoid biosynthesis pathway, and the compounds synthesized in this pathway were mainly phenolics [35,36,37]. Fifteen upregulated genes were annotated in the phenylpropanoid biosynthesis pathway, and three, three and five upregulated genes were annotated to the chlorogenic acid, flavonoid, and lignin biosynthetic pathways, respectively. *PAL* (phenylalanine ammonia-lyase) was annotated to the upregulated gene, which is the first enzyme of the phenylpropane metabolic pathway and a key enzyme in the biosynthesis of numerous secondary metabolites [38,39,40]. *PAL, 4CL* (4-Coumarate: coenzyme A ligase) and *HCT* (hydroxycinnamoyl-CoA shikimate), annotated to the upregulated genes, are key enzymes for the synthesis of chlorogenic acid, a class of enzymes that are produced from caffeic acid and quinic acid, which are phenolic compounds [41]. Several studies have shown that chlorogenic acid has powerful radical scavenging ability [42,43,44], and that sweetpotato leaves contain large amounts of chlorogenic acid [5,45]. Therefore chlorogenic acid would most likely be an important active component in the antioxidants of sweetpotato blades. However, chlorogenic acid is a class of polyphenolic compounds, and further studies are needed to determine which specific monomeric chlorogenic acids are responsible for the antioxidant activity. Other key genes involved in flavonoid biosynthesis were also annotated into upregulated genes, such as: *CHS* (Chalcone synthase), *CHI* (Chalcone isomerase), *F3’H* (Flavanone 3-hydroxylase). Flavonoids, another important class of polyphenolic compounds in sweetpotato blades, also have a strong ability to scavenge free radicals. The flavonoid biosynthesis pathway is relatively conserved in plants [46], and many flavonoid synthesis-related genes of sweetpotato, such as: *IbCHS* [47], *IbCHI* [48], and *IbF3’H* [49], have been cloned from sweetpotato storage roots. In addition, the lignin synthesis-related genes *CCoAOMT* (caffeoyl-CoAO- methyltransferase), *POD* (peroxidase), *CAD* (cinnamyl-alcohol dehydrogenase), and *F5’H* (ferulic acid 5-hydroxylase) were annotated to upregulated genes. Lignin is composed of phenylpropanoid structural units, with active groups such as: phenolic hydroxyl, alcohol hydroxyl, carbonyl, methoxy, carboxyl and conjugated double bonds in its molecular structure [50], and studies have shown that lignin also has a strong antioxidant activity. Gong et al. [51] investigated the antioxidant activity of lignin extracted from bamboo and found that the antioxidant activity of lignin was stronger than that of the synthetic antioxidant, BHT (butylated hydroxytoluene). Studies on sweetpotato lignin mainly focus on disease resistance and healing tissue properties, and the relationship between lignin and antioxidant in sweetpotato blades has not been reported. Thus, further study is still needed in the future.

As a polyploid crop with a complex genome, sweetpotatoes of different genotypes have great differences. Differences among genotypes may lead to differences in antioxidant activity in sweetpotato blades. Our study showed that genotypes with high antioxidant activity have high TPC and TFC; DEGs annotated the phenylpropanoid biosynthesis pathway as the main differential metabolic pathway, and the differences in biosynthesis of chlorogenic acid, flavonoids and lignin may be the main factors for the antioxidant differences in sweetpotato blades. Breeding of sweetpotato varieties is a standardized process. In order to realize the utilization of sweetpotato blades with high phenolics and antioxidant activity, subsequently, we will evaluate antinutritional factors, agronomic traits and food sensory. This study is an important guide for the selection and breeding of varieties with high antioxidant activity and polyphenols in sweetpotato blades.

## 5. Conclusions

In this study, we determined the ABTS+, DPPH·, FRAP, TPC and TFC of 23 genotypes of sweetpotato blades and screened out genotypes such as ‘Wan1314-6’, ‘Yushu1’ and ‘18-2-4’ with high antioxidant activity and phenolics in the blades. Correlation analysis showed a positive correlation between ABTS+, DPPH, FRAP, TPC and TFC, indicating that TPC and TFC have positive effects on the antioxidant activity of sweetpotato blades. Analysis of transcriptome data from two genotypes, ‘Shangshu19’ and ‘Wan1314-6’, showed that the phenylpropanoid biosynthetic pathway was responsible for the differences in antioxidant activity in sweetpotato blades of different genotypes. In addition, 15 upregulated genes were annotated for metabolic pathways, which were mostly annotated in the chlorogenic acid, flavonoid and lignin biosynthesis pathway.

## Figures and Tables

**Figure 1 genes-13-01078-f001:**
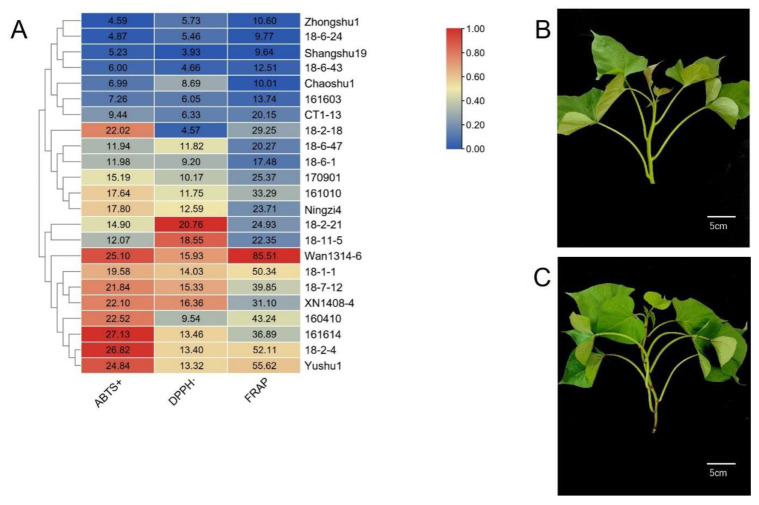
(**A**) Heatmap of antioxidant activity of blades from 23 sweetpotato genotypes. (**B**) Low antioxidant activity genotype ‘Shangshu 19’. (**C**) High antioxidant activity genotype ‘Wan1314-6’. **Note:** The values in Figure (**A**) are the measured antioxidant activity values. Heatmap production with reference to the maximum value of each column. In the figure, ‘Shangshu19’ and ‘Wan1314-6’ are used for RNA-seq. ABTS+: 2’-azinobis-(3-ethylbenzothiazoline-6-sulphonic acid) free radicals; DPPH·: 2,2-diphenyl-1-picrylhydrazyl; FRAP: Ferric ion reducing antioxidant power.

**Figure 2 genes-13-01078-f002:**
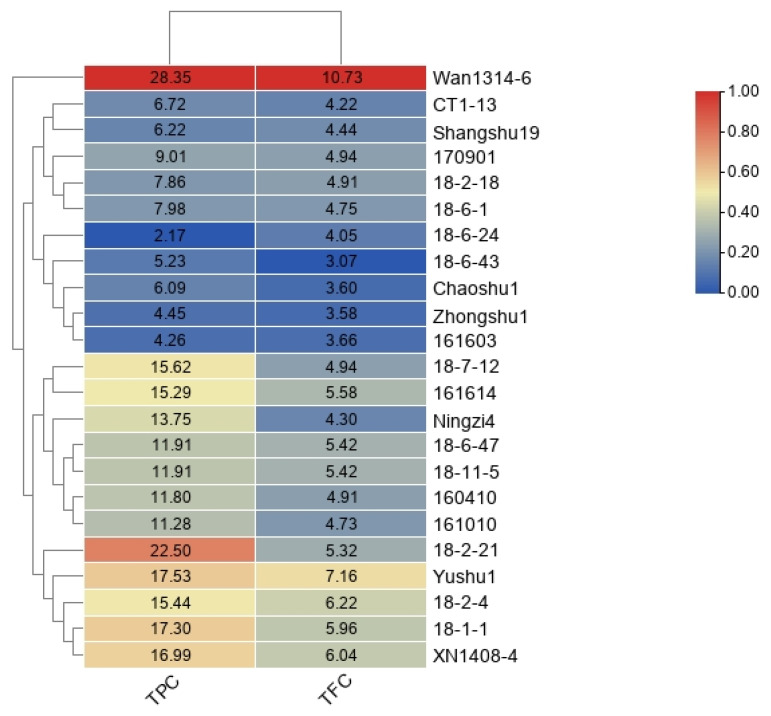
Heatmap of TPC and TFC of blades from 23 sweetpotato genotypes. **Note:** The values in the figure are the measured TPC or TFC values. Heatmap production with reference to the maximum value of each column. TPC: total polyphenol content; TFC: total flavonoid content.

**Figure 3 genes-13-01078-f003:**
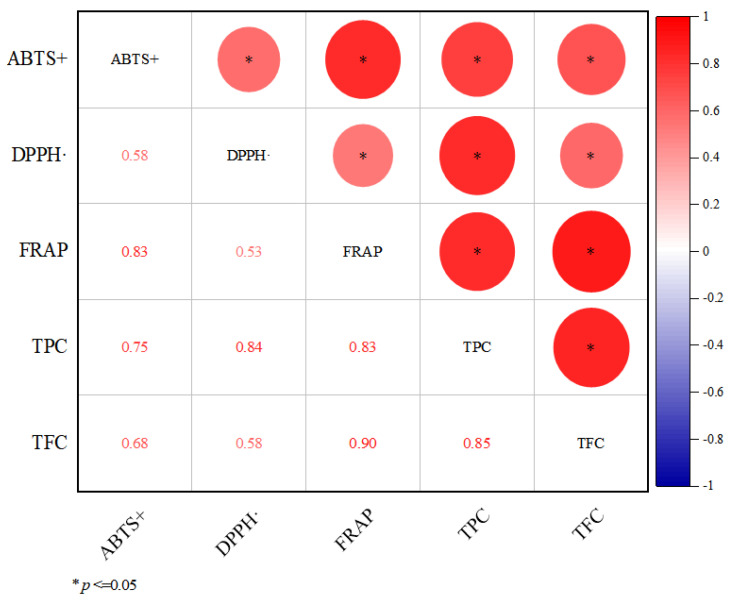
Correlation analysis of TPC, TFC and antioxidant activity of sweetpotato blades. **Note:** The circle size represents the correlation size. The shade of red represents the size of the positive correlation.

**Figure 4 genes-13-01078-f004:**
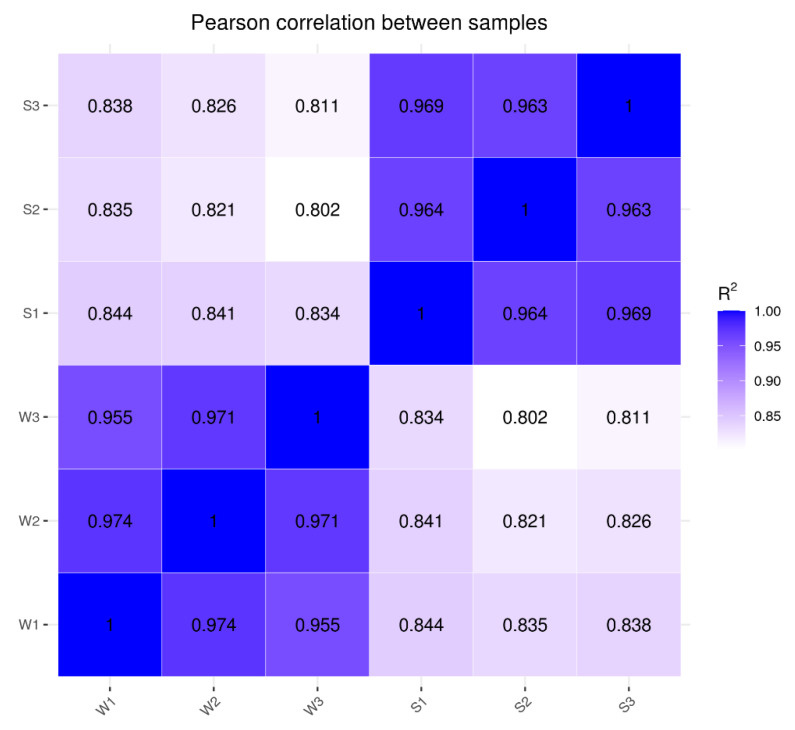
Heatmap of correlation coefficient. **Note:** W: Wan1314-6, S: Shangshu19.

**Figure 5 genes-13-01078-f005:**
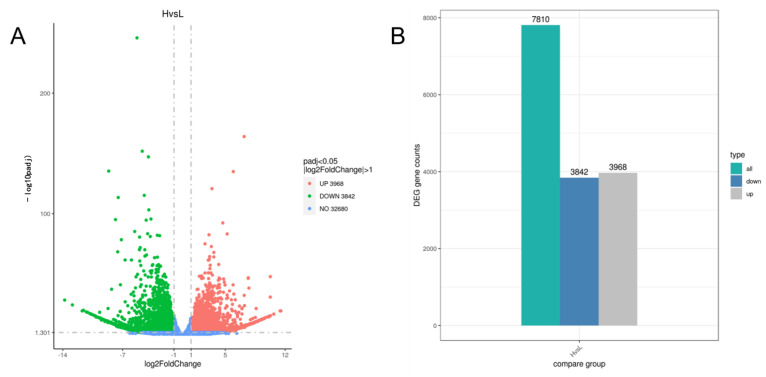
DEGs distribution. (**A**) DEGs volcano map. (**B**) DEGs column chart. **Note:** all: all the DEGs; up: upregulated DEGs; down: downregulated DEGs.

**Figure 6 genes-13-01078-f006:**
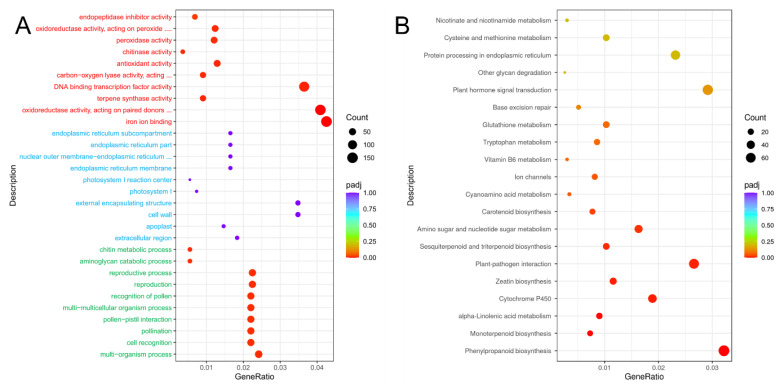
GO enrichment analysis and KEGG metabolic pathway analysis of DEGs. (**A**) GO enrichment analysis scatter plot. (**B**) KEGG metabolic pathway analysis scatter diagram. **Note:** green font indicates ‘biological process’; blue font indicates ‘cellular component’; red font indicates ‘molecular function’.

**Figure 7 genes-13-01078-f007:**
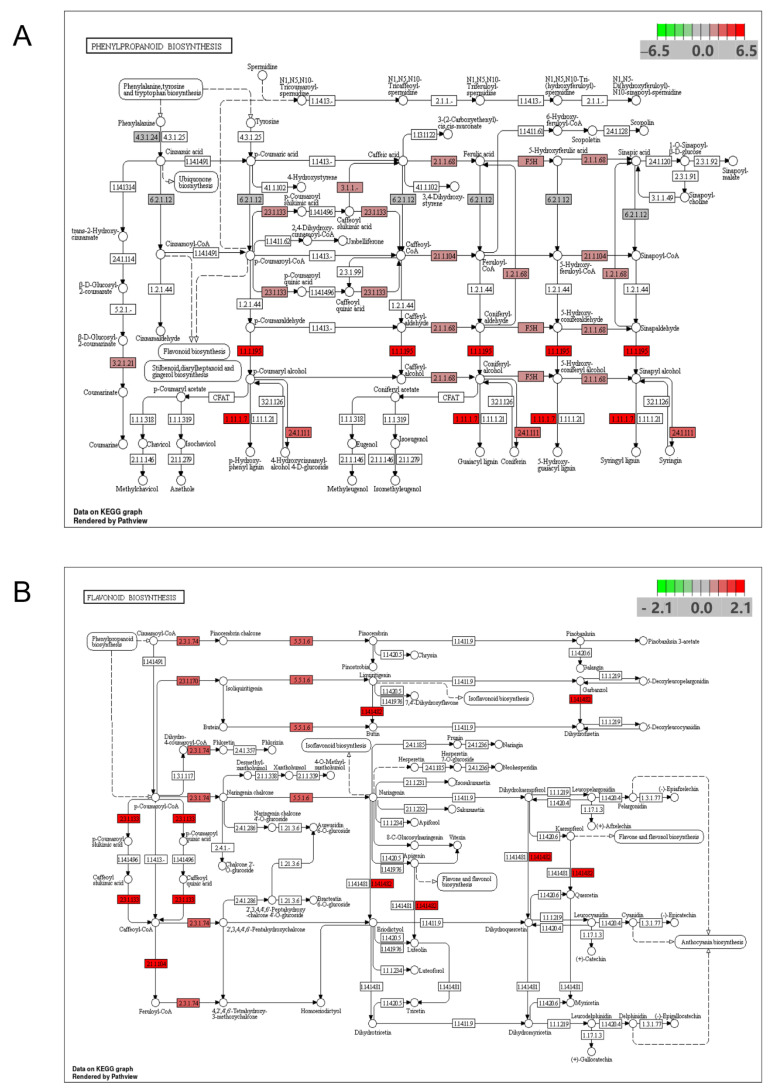
Genes upregulated in the KEGG pathway are designated in red. (**A**) Biosynthetic pathway of phenylalanine. (**B**) Biosynthetic pathway of flavonoid.

**Figure 8 genes-13-01078-f008:**
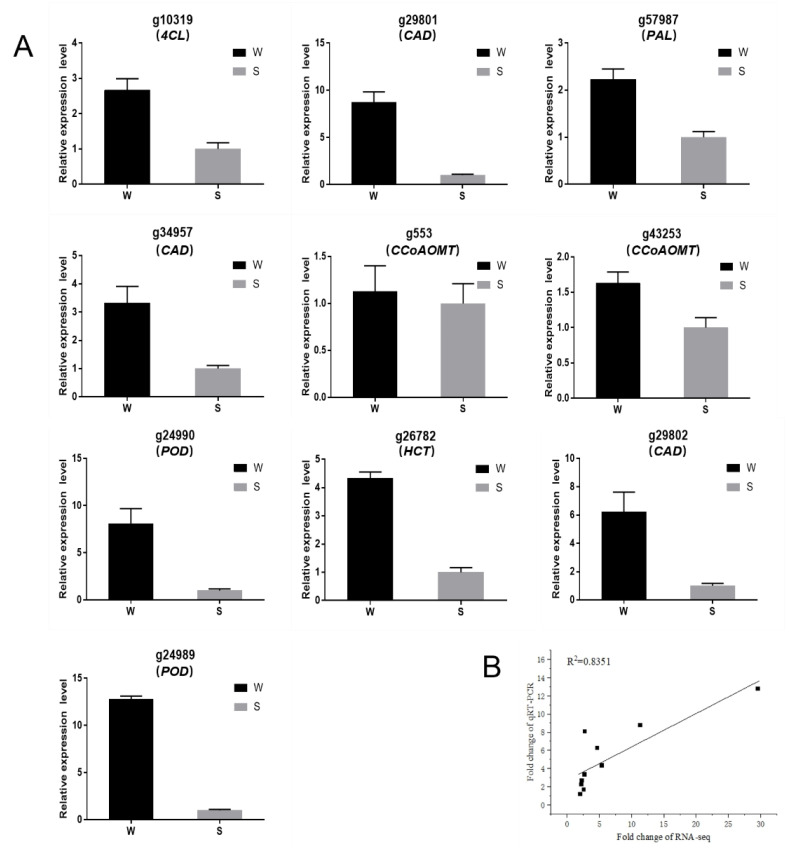
The qRT-PCR of differentially expressed genes. (**A**) Expression of 10 differential genes in two genotypes of sweetpotato blades. (**B**) Correlation analysis of qRT-PCR and RNA-seq. **Note:** W: Wan1314-6; S: Shangshu19.

**Table 1 genes-13-01078-t001:** Evaluation of the quality of sweetpotato Wan1314-6 and Shangshu19 blades sequencing data.

	Raw Reads	Clean Reads	Clean Bases	Error Rate	Q20	Q30	GC Content
W1	47094680	44240870	6.64G	0.03	97.57	93.45	46.59
W2	45400034	44564352	6.68G	0.03	96.99	92.00	46.13
W3	43525644	41821630	6.27G	0.03	97.66	93.62	46.27
S1	42353168	40573440	6.09G	0.03	97.65	93.60	46.48
S2	42027724	40358894	6.05G	0.03	97.67	93.64	46.33
S3	44622468	42746376	6.41G	0.03	97.61	93.46	46.37

**Note:** W: Wanshu1314-6; S: Shangshu19.

## Data Availability

The raw sequence data of the current study were deposited in the NCBI Sequence Read Archive (SRA), http://www.ncbi.nlm.nih.gov/bioproject/PRJNA834574 (accessed on 6 May 2022).

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
