# Peer review of "Antioxidant Activity of Phenolic Extraction from Different Sweetpotato (Ipomoea batatas (L.) Lam.) Blades and Comparative Transcriptome Analysis Reveals Differentially Expressed Genes of Phenolic Metabolism in Two Genotypes"

_genes, 2022, doi:10.3390/genes13061078_

Round 1

Reviewer 1 Report

The paper brought new information to progress toward sweet potato polyphenols and antioxidant activity screening  but to much work still needed to assist breeding of this crop and valuing blades . Does there any evidence of antinutritional factors or toxicity of the plant parts that author want to make it valuable ?

What about polyphenol level heritability is it under high GXE effect ? 

Reviewer 2 Report

I’m delighted to review the manuscript entitled “Antioxidant activity of phenolic extraction from different sweetpotato [Ipomoea batatas (L.) Lam.] blade and comparative transcriptome reveal differentially expressed genes of phenolic metabolism in two genotypes”

The manuscript deals with the probable molecular mechanism behind the phenolic compounds metabolism in the leaf blade of sweetpotato. The work has started with the analysis of the antioxidant activity, total phenolic, and total flavonoid contents of a multiple sweetpotato blades genotypes. Followed by the analysis RNA-sequencing of 2 genotypes that demonstrates important differences. And finish by concluding that ‘Wan1314-6’, ‘Yushu1’ 259, and ‘18-2-4’ with high contents of TPC and TFC and strong antioxidant activity would be suitable for breeding sweetpotato varieties with high antioxidant activity in their blade. This is very interesting research that select sweetpotato genotypes with high antioxidant activity and high phenolic compounds content in their leaf blade.

The manuscript is written in clear language, and pertinent references enforce the data shown in the manuscript. The material and methods sections are explained properly. Results are interpreted reasonably and the study issues are discussed appropriately. At the end of the article, the authors have made engaging and correct conclusions. The article, however, requires revision on the following points:

-Lines 54-56: unclear sentence need to be rewrite. 

-Line 61: Please add “ABA” meaning.

-Lines 68-69: the authors write a very strange phrase “The studies about molecular mechanism of the phenolic metabolism in sweetpotato blade is little” If there is studies a few studies about molecular mechanism of the phenolic metabolism in sweetpotato blade, I invite the authors to include those studies in this paragraph.

-Line 161: “3.07” instead of “3.58”

-Line 162: “18-6-43” not “Zhongshu1”

The figures should be as clear as possible, please add abbreviations meaning in the figure legend.
